# Large-Scale Drug Screen Identifies FDA-Approved Drugs for Repurposing in Sickle-Cell Disease

**DOI:** 10.3390/jcm9072276

**Published:** 2020-07-17

**Authors:** Matthew Cannon, Hannah Phillips, Sidney Smith, Katie Williams, Lindsey Brinton, Charles Gregory, Kristina Landes, Payal Desai, John Byrd, Rosa Lapalombella

**Affiliations:** 1Division of Hematology, The Ohio State University, Columbus, OH 43210, USA; Matthew.Cannon@osumc.edu (M.C.); Hannah.Phillips@osumc.edu (H.P.); Sidney.Smith@osumc.edu (S.S.); Katie.Williams@osumc.edu (K.W.); Lindsey.Brinton@osumc.edu (L.B.); Charles.Gregory@osumc.edu (C.G.); Kristina.Landes@osumc.edu (K.L.); Payal.Desai@osumc.edu (P.D.); John.Byrd@osumc.edu (J.B.); 2College of Veterinary Medicine, The Ohio State University, Columbus, OH 43210, USA; 3Division of Pharmaceutics, College of Pharmacy, The Ohio State University, Columbus, OH 43210, USA

**Keywords:** sickle-cell disease, NIH clinical collection, fetal hemoglobin, drug screen, in-cell western, drug repurposing

## Abstract

Sickle-cell disease (SCD) is a debilitating hematological disorder with very few approved treatment options. Therapeutic reactivation of fetal hemoglobin (HbF) is one of the most pursued methods for ameliorating the systemic manifestations of SCD. Despite this, very few pharmacological agents have advanced to clinical trials or marketing for use. In this study, we report the development of an HbF in situ intracellular immunoblot assay coupled to a high-throughput drug screen to identify Food and Drug Administration (FDA) approved drugs that can be repurposed clinically for treatment of SCD. Using this assay we evaluated the National Institute of Health (NIH) Clinical Collection (NCC), a publicly available library of 725 small molecules, and found nine candidates that can significantly re-express HbF in erythroid cell lines as well as primary erythroblasts derived from SCD patients. Furthermore, we show the strong effects on HbF expression of these candidates to occur with minimal cytotoxicity in 7 of the 9 drugs. Given these data and their proven history of use for other indications, we hypothesize that several of these candidate drugs warrant further investigation for use in SCD.

## 1. Introduction

Sickle-cell disease (SCD) is a debilitating genetic disorder that affects millions of people worldwide. The disorder is the result of a point mutation on chromosome 11 in codon 6 of the beta-globin gene locus that changes a GAG codon to a GTG. This results in a single amino acid substitution from a hydrophilic glutamate to a hydrophobic valine. The resulting, mutated beta-globin protein (β^S^) in the setting of hypoxia undergoes polymerization with alpha-globin proteins (α) to form aberrant hemoglobin polymers (HbS). These polymers retain different biochemical properties than their normal adult hemoglobin (HbA) counterparts that lead to a cascade of negative effects on the patient’s pathophysiology of tissue circulation and oxygenation [1]. When deoxygenated, HbS polyermizes into long chains that deform the red cell’s shape into the disease’s characteristic sickle namesake. Once sickled, affected RBCs are prone to binding to each other and the endothelium, which in turn blocks blood flow and causes severe pain as a direct consequence of ischemic injury to the surrounding tissue and organs. This event is known as a vaso-occlusive crisis (VOC) and results in a systemic response in which the patient remains in a state of persistent, diffuse vascular inflammation [2,3]. This systemic response further upregulates inflammatory cytokine signaling and promotes increased adhesion molecule expression on the endothelium, thus making the patient prone to additional VOCs. In addition, sickle RBCs have increased fragility and a decreased lifespan, making them prone to lysis. Increased lysis of RBCs causes a systemic increase in circulating free hemoglobin (Hb) and reactive oxygen species (ROS), both of which further promote inflammation and contribute to damage in multiple organs. Accordingly, sickle-cell patients enter a vicious cycle of VOCs and inflammation that drive further chronic vascular injury. Despite the severity of the disease and these symptoms, SCD has had very few treatment options to date.

Until recently, hydroxyurea represented the only pharmacologic intervention strategy to treat SCD. Initially approved in 1997 due to its ability to reduce VOC and transfusion frequency, it was later found to attack SCD pathology at multiple fronts through its ability to simultaneously increase fetal hemoglobin (HbF) and decrease neutrophils [4]. Increases in circulating HbF content are known to ameliorate the SCD phenotype through the ability of the gamma-globin subunit to disrupt HbS polymer chaining when present, thus protecting against sickling and VOCs. No new pharmacologic agents were approved until 2017, when L-glutamine was approved due to its ability to protect against pro-inflammatory reactive oxygen species (ROS) and reduce yearly VOCs [5]. Following the success of L-glutamine, crizanlizumab and voxelotor were both approved for use in SCD in the fall of 2019 [6,7,8,9]. Crizanlizumab is a P-selectin inhibitor that acts to block sickle-cell adhesion to the endothelium, thus preventing the incidence of VOCs by 45%. Voxelotor, formerly called GBT440, acts a hemoglobin modulator that is able to directly bind to the alpha-globin chain of HbS polymers and allosterically stabilize the oxygenated state of the molecule, thus inhibiting HbS chain polymerization, reducing hemolysis, and increasing patient hemoglobin levels. More recently, hopes for the disease include genetic editing through clustered regularly interspaced short palindromic repeats (CRISPR)-based systems [10,11]. While preliminary results are encouraging, the use of genetic editing in humans to treat diseases is still in its infancy and the long-term side-effects of such efforts remains to be seen. Additionally, the economic costs of recent therapeutic efforts can be a significant barrier for some (>$2000 per month) [8,12]. Furthermore, the economic challenge of bringing these new treatments to other nations of the world where SCD is more prevalent represents a significant barrier on its own.

At present there is still a great need for readily available, economically relevant SCD therapies that can be administered worldwide with acceptable toxicity.

Clinical research in sickle-cell anemia has focused on several areas including the pharmacological re-expression of a key hemoglobin polymer named fetal hemoglobin (HbF) [13,14]. Fetal hemoglobin is a naturally occurring hemoglobin molecule that is present throughout fetal development. In most children, hemoglobin F decreases and is replaced with hemoglobin A during the first few months of life except in the presence of a rare benign asymptomatic genetic disorder, hereditary persistence of fetal hemoglobin (HPFH). The result of HPFH is an asymptomatic, modest elevation of HbF. Patients with SCD who possess the HPFH hereditary trait tend to have a higher quality of life and experience less frequent hospitalizations compared to HPFH negative SCD patients [15,16]. This occurs due to the protective effects of HbF against HbS. Specifically, HbF blocks HbS from dimerizing, thereby preventing hemoglobin polymer chaining and decreasing the probability of a vaso-occlusion [1,17]. Hydroxyurea, thalidomide, sodium butyrate, and decitabine, a DNMT1 targeting agent, have been previously shown to work through induction of HbF and subsequent protection against VOCs [5,18,19,20,21,22].

Given the long-term success of hydroxyurea in treating SCD, we focused efforts on developing an in situ HbF intracellular immunoblot assay coupled to a high-throughput drug screen to identify drugs that can be repurposed for treatment of SCD. This is a powerful, yet previously under-utilized methodology to screen for modulation of HbF from a library of drugs that have already been FDA approved for other indications. Through this process, known as drug repurposing, we have identified existing candidate drugs for use in SCD, thus drastically reducing the cost of development for new drugs to market and allowing for fast track potential of old drugs for new indications [23]. In this report, we present nine new drugs for repurposing in SCD that warrant future clinical investigation.

## 2. Experimental Section

### 2.1. Cell Culture

Cell lines K562 (isolated from 53 y.o. female, CML), THP-1 (isolated from 1 y.o. male, AML), and HEL92.1.7 (isolated from 30 y.o. male, erythroleukemia) suspension cell lines were purchased from the American Type Culture Collection (ATCC, Manassas, VA, USA) and cultured in Roswell Park Memorial Institute (RPMI) media (Invitrogen, Grand Island, NY, USA) supplemented with 10% fetal bovine serum, 10,000 units of penicillin, 10 mg of streptomycin and 200 mM of glutamate. All cells were kept in a 37 °C, 5% CO_2_ incubator.

Blood was obtained from healthy volunteers and sickle cell donors following written informed consent under a protocol approved by the Institutional Review Board (IRB) of The Ohio State University (OSU; Columbus, OH, USA) (IRB Protocol Number: 1997C0194) in accordance with the Declaration of Helsinki. Blood was used from across 42 consenting sickle-cell donor volunteers to develop and perform the experiments used across this study. Peripheral blood mononuclear cells (PBMCs) were obtained using ficoll centrifugation. PBMCs were cultured under a consecutive two-phase culture system: (I) RPMI supplemented with 10% FBS, Stem Cell Factor (10 ng/uL, Sigma-Aldrich #H8416), Erythropoietin (1.5 U/uL, Sigma-Aldrich #H5166), Interleukin-3 (1 ng/uL, Sigma-Aldrich #H7166), and Dexamethasone (1 uM, Selleck Chem #S4028) for one week and (II) MethoCult^TM^ Optimum (#4034) following standard colony plating assay conditions for an additional week.

### 2.2. In-Cell Western Blot

Cell lines (K562, THP-1, and HEL92.1.7) and in vitro differentiated RBCs were cultured in black 96-well or 384-well plates in RPMI media (Invitrogen, Grand Island, NY, USA) supplemented with 10% FBS, 10,000 units of penicillin, 10 mg of streptomycin and 200 mM of glutamate. All cells were kept in a 37 °C, 5% CO_2_ incubator. After incubation, plates were spun down (1500 rpm, 5 min) and supernatant was removed without disturbing the cell layer. Cells were then fixed in-plate using 3.7% formaldehyde for 20 min. After fixation, standard in-cell Western assays were performed as previously published by LI-COR Biosciences.

### 2.3. National Institute of Health (NIH) Clinical Collection (NCC) Screen

The NIH Clinical Collection (NCC) is a library containing 725 small molecules and inhibitors that was made public via Common Fund support as a part of the Molecular Libraries and Imaging program (http://commonfund.nih.gov/molecularlibraries/tools). K562 and in vitro differentiated RBC cells were plated in black 384-well plates using a MultiDrop Combi Reagent Dispenser (ThermoFisher Scientific, Waltham, MA, USA, 5) at 5 e^3^ and 2.5 e^4^ cells per well respectively. The NCC library was dispensed across the plates by a BioMek Automated Workstation. All cells were kept in a 37 °C, 5% CO_2_ incubator. After incubation, plates were spun down (1500 rpm, 5 min) and supernatant was removed without disturbing the cell layer. Cells were then fixed in-plate using 3.7% formaldehyde for 20 min. After fixation, standard in-cell Western assays were performed as previously published by LICOR.

### 2.4. MTS Viability Assays

K562 and HEL92.1.7 cell lines were cultured in 96-well plates at 50,000 cells per well in RPMI media (Invitrogen, Grand Island, NY, USA) supplemented with 10% FBS, 10,000 units of penicillin, 10 mg of streptomycin and 200 mM of glutamate. After incubation, a solution of MTS tetrazolium (3-(4,5-dimethylthiazol-2-yl)-5-(3-carboxymethoxyphenyl)-2-(4-sulfophenyl)-2H-tetrazolium)) Promega, G1111) and phenazine methosulfate (Sigma-Aldrich, St. Louis, MO, P9625) was added to all plate wells. Plates were then incubated for an additional 3 h and then measured via plate reader at 490 nm absorbance.

### 2.5. Immunoblot

Cells were lysed in standard lysis buffer (10 mM Tris pH 7.4, 150 mM NaCl, 1% Triton X-100, 1% deoxycholic acid, 0.1% SDS, 5 mM ethylenediamine tetraacetic acid supplemented with phosphatase inhibitor cocktail II (Sigma-Aldrich, Madison, WI, USA, P2850), phosphatase inhibitor cocktail III (Sigma-Aldrich, P5726), protease inhibitor cocktail Σ (Sigma-Aldrich, P-8340), and phenylmethyl sulfonyl fluoride (Sigma-Aldrich, P-7626). Protein concentrations were determined using bicinchoninic acid assay (BCA). Lysates were run on sodium dodecyl sulfate-polyacrylamide gels, transferred onto nitrocellulose membranes and incubated with antibody. Membranes were developed using SuperSignal West Pico PLUS Chemiluminescent Substrate (Thermofisher Scientific, 34580).

### 2.6. Circular Chromosome Conformation Capture (4C)

K562 and HEL92.1.7 cell lines were grown in culture under treatment for 72 h and lysed for DNA collection. 4C libraries were generated using previously published procedures and used DpnII and Csp6I for subsequent RE digestions. 4C primers were designed for the HBG2 promoter region (FW: CAAAGCACCTGGATGATC, RV: TTGTCTCTAGCTCCAGTGAG). Libraries were sequenced on an Illumina MiSeq system and analyzed using 4Cseq analysis pipeline. 

### 2.7. Statistical Analysis

Data are shown as the mean ± standard error of the mean (SEM). The significance of difference in antibody ratios was evaluated by two-tailed, unpaired Student’s t-test. All statistical procedures were performed using commercial software (GraphPad Prism, version 8.4.2, GraphPad Software Inc., San Diego, CA, USA).

## 3. Results

### 3.1. Fetal Hemoglobin Expression on In-Cell Western Correlates with Traditional Western Blot Analysis

We first sought to establish the in-cell Western technique with our HbF antibody and cell line conditions. K562 and THP-1 cell lines were plated in 10-fold dilutions from 1 e6 cells per well to 1 e4 cells per well in a black 96-well plate. Cells were then fixed and a standard in-cell Western assay was performed (Figure 1A). Half-step dilutions starting at 1:50 of HbF antibody were tested across each cell dilution. Upon completion plates were scanned and background-corrected antibody signal ratios were calculated for each well. The antibody signal ratios of HbF-positive K562 cell line was then compared against HbF-negative THP-1 cell line to gauge the specificity of the antibody. This analysis showed a 12-fold stronger antibody signal in K562 cell line at the most concentrated conditions (*p* = 0.0049) (Figure 1B). Additionally, when plated at 1 e^5^ cells per well, K562 cell line still showed greater antibody signal than HbF-negative THP-1 cell line plated at their most concentrated conditions, further suggesting that the HbF antibody is indeed binding to the correct target.

We next sought to test the achievable limits of detection of this assay. K562, HEL92.1.7 and THP-1 cell lines were plated in 10-fold dilutions from 1 e^5^ cells per well to 1 e^2^ cells per well in a black 96-well plate. As prior, cells were fixed and a standard in-cell Western assay was performed. Upon completion plates were scanned and background corrected antibody signal ratios were calculated for each well. Next, the antibody signal ratios of HbF-positive K562 cell line was compared against HbF-negative THP-1 cell line. This analysis showed a large difference in antibody signal in K562 cell line at their most concentrated conditions and, as previously, lower concentrations of K562 cells showed stronger signal than HbF-negative THP-1 cells at their highest concentration (Appendix A).

We next sought to cross-validate the results of in-cell Westerns to the traditional Western blot. To accomplish this, we treated in vitro differentiated RBCs from normal donors with three experimental LSD1 inhibitors: ORY-1001, GSK-LSD1 and GSK-2479 that previously have been shown to up-regulate fetal hemoglobin [24,25]. Inhibition of LSD1 is a proven mechanism for re-expression of fetal hemoglobin that has seen pre-clinical success in mouse and baboon animal models [26,27,28]. Cells were treated for 48 h with LSD1 inhibition and subject to both in-cell Western and traditional immunoblot. Relative to their respective DMSO conditions, ORY-1001 and GSK-LSD1 had similar increases of HbF expression compared to traditional Western blot (Appendix A). Together, these data validate our HbF antibody in the in-cell Western technique and highlight its sensitivity to changes in HbF expression with as a few as 1000 cells.

### 3.2. Sequential Drug Screens Show New Candidates for Sickle-Cell Disease (SCD)

Next, we sought to use our newly developed in-cell Western assay to screen the NIH Clinical Collection (NCC, http://commonfund.nih.gov/molecularlibraries/tools) for modulators of fetal hemoglobin. K562 cell line were plated in 384-well plates at 5 e^3^ cells per well using a MultiDrop Combi Reagent Dispenser. The NCC and DMSO controls were added in duplicate across the plates using a BioMek Automated Workstation (Figure 2A). After 3 days incubation, all cells were fixed and a standard in-cell Western was performed on each plate. Plates were scanned and background corrected antibody signal ratios were calculated for each well. Signal ratios for each duplicate were compared against DMSO controls for increased expression of HbF. Cytarabine was used a positive control for induction of HbF in K562 cells based upon previous published works [29,30]. While on average test drugs had no significant increase in HbF, individual examination of each plate revealed 31 drugs with at least 1.5-fold higher expression of HbF as compared to DMSO controls (Figure 2B). Among these drugs were cytotoxic agents (such as epirubicin and daunorubicin), targeted therapy (imatinib mesylate), and anti-infectious agents (such as pyrimethamine, quinidine hydrochloride and mafenide acetate). Twenty drugs with the highest relative fold-change are shown in Table 1.

Following up on these results, we repeated the screen using CD45+ cells isolated from a sickle-cell patient who had been in vitro differentiated towards an erythroid lineage as described in the methods (referred to here as iRBC). The in-cell Western experiment was repeated using iRBCs plated at 2.5 e^4^ cells per well. From this screen we found 103 drugs with at least 1.5-fold higher expression HbF than DMSO. Twenty drugs with the highest relative fold-change are shown in Table 2. Comparing the results of these two screens reveals an overlap of nine drugs with at least a 1.5-fold higher expression of HbF in both K562 s and iRBCs. These drugs are: daunorubicin, epirubicin, pyrimethamine, mafenide acetate, prednisolone, quinidine hydrochloride, perphenazine, felodipine and duvadilan. These drugs and their fold-changes are shown in Table 3. Together these screen results highlight the existence of drugs with prior FDA approval that can increase HbF expression in human cells.

### 3.3. Validation and Cytotoxicity Evaluation of NCC Screen Results

After identifying the top candidates from our overlapping screens, we next sought to evaluate each individual molecule for their effects on HbF expression in K562 and HEL92.1.7 cell lines. Cells were cultured with each candidate drug and allowed to incubate. After 72 h, lysates were collected and subject to immunoblot analysis for HbF. This analysis revealed differing levels of HbF re-expression by candidate molecules in K562 and HEL92.1.7 cell lines (Figure 3A). Pyrimethamine, imatinib, quinidine hydrochloride, felodipine and mafenide acetate increased HbF expression in K562 cells after 72 h. In HEL92.1.7 cells, however, all tested agents increased the expression of HbF. Pyrimethamine, imatinib and quinidine hydrochloride seemed to have the strongest and most consistent effects on increasing HbF expression in the two cell lines.

Additionally, although these drugs are already FDA approved, we also examined their cytotoxicity profile via MTS assay. Both cell lines were incubated with each candidate drug for 72 h. MTS reagent was added and absorbance was read. This analysis revealed a lack of cytotoxicity across most compounds (Figure 3B). Notable exceptions to this include daunorubicin and epirubicin. Daunorubicin and epirubicin are anthracyclines commonly used in chemotherapy and thus their toxicity on these cell lines is expected. It is important to note that since these drugs are administered intravenously and commonly are associated with adverse side effects, it is highly unlikely that these drugs would translate well to SCD patients. One additional exception is imatinib. Imatinib (Gleevec) was originally developed to target the *Bcr-Abl* gene fusion protein driving chronic myeloid leukemia (CML). The K562 cell line contain this gene fusion protein but HEL92.1.7 s do not and this is reflected aptly in their cytotoxicity profiles (Figure 3B). Despite this difference in Bcr-Abl kinase expression it is interesting to note the increase in HbF occurs in both cell lines suggesting this is independent of BCR-ABL fusion protein inhibition. Imatinib is known to bind other additional tyrosine kinase receptors, notably c-Kit, platelet-derived growth factor receptor (PDGFR), and c-Abl kinase receptors. Of these targets, c-Kit has been shown previously to play a large role in sustaining proliferation and delaying terminal maturation, particularly in erythroid progenitors. In these cells, c-Kit signaling downregulates the expression of erythroid signaling and transcription factors, notably master regulators GATA-1 and KLF1 [31,32]. Given that GATA-1 positively modulates HbF expression via gamma-globin promoter binding, we hypothesize that the increase in HbF protein seen is occurring as a result of inhibited c-Kit signaling [33].

### 3.4. Changes in Cis Interactions at the HBG2 Promoter Induced by Imatinib in K562 and HEL92.1.7 Cell Lines

Given the drastic increase in HbF expression with imatinib treatment in our cell lines, we next investigated any genetic changes that may be occurring via chromosome conformation capture-on-chip (4C). As in previous experiments, K562 and HEL92.1.7 cell lines were cultured with imatinib for 72 h before being collected and lysed for the 4C pipeline. After subsequent ligation and digestions, DNA was sequenced and analyzed for *cis* interactions on chromosome 11 using the HBG2 promoter region as bait.

Analysis of K562 4C peaks shows the HBG2 promoter region to be interacting with several olfactory receptor genes upstream and downstream of HBG2 at chromosomal regions 5.15, 5.17, 5.22 and 5.31 (Figure 4A). Interestingly, treatment with imatinib did not seem to affect where the HBG2 promoter region interacts but instead seemed to increase the frequency of these interactions, especially at chromosomal region 5.17. In comparison, analysis of HEL92.1.7 4C peaks shows the HBG2 promoter region to retain many of the same *cis* interactions seen in K562 cells, but also many unique peaks like those at chromosomal regions 5.11 and 5.34 (Figure 4B). Unlike in K562 cells, treatment of HELs with imatinib seemed to drastically alter the *cis* interactions of the HBG2 promoter region, including the disappearance of both unique peaks at 5.11 and 5.34. The resulting changes in *cis* interactions leaves a peak profile more akin to the HbF-high expression K562 cell line, suggesting these regions to be important for regulation of HbF expression. Taken together, these results suggest that imatinib is able to induce changes in chromatin arrangement that increase HbF expression.

## 4. Discussion

The present study is one of the first to screen a large library of FDA-approved drugs for repurposing to SCD. In our technical testing, we showed the previously underutilized in-cell Western to be a powerful and sensitive technique for screening for modulators of fetal hemoglobin. Sequential, large-scale screens on K562 cells and in vitro differentiated red blood cells revealed an overlap of 9 drugs that can positively modulate HbF expression. Importantly, all nine drug hits have FDA approval for various indications and have promise for repurposing therapeutically toward SCD. Furthermore, individual validation experiments confirmed the ability of these drugs to increase HbF expression with little to no effects on cytotoxicity. Of the drugs validated in this study, imatinib, pyrimethamine, and quinidine show the strongest effects on HbF expression. Additional pre-clinical studies are underway in our laboratory to further characterize these drugs for use in SCD.

The in-cell western methodology used in this study remains a powerful yet underutilized tool for large-scale studies. We developed this assay to detect intracellular HbF and then combined it with a large-scale drug screen utilizing the NIH Clinical Collection (NCC), a plated array of small molecules with a history of use in humans and prior FDA approval. While this collection was originally assembled as part of the Molecular Libraries and Imaging Initiative to promote large screen experiments, it has previously never been used in the field of SCD. With the recent push for new therapeutic options to treat SCD, high powered detection assays such as ours are invaluable in finding new drugs. Furthermore, one main advantage our assay retains over others is the ease of translation to other libraries. While our work focused solely on the NCC, other drug libraries and small molecule collections can just as easily be plated and tested for positive modulators of HbF. Lastly, it is important to note that the K562 cell line used in our initial screen can be chemically stimulated towards erythroid differentiation where accumulation of HbF naturally occurs. Accordingly, future experiments using this methodology should utilize primary cells for increased reliability and analysis power. While genetic editing and other treatments are in development for SCD, our methodology represents a new way to quickly identify new or existing pharmacologic agents that can be quickly fast-tracked to fight the disease.

In our experiments we identified nine drugs as positive modulators of HbF. Of these drug candidates, we showed pyrimethamine, mafenide acetate, and quinidine hydrochloride to have strong effects on HbF expression in our model cell lines. Interestingly, all three of these drugs have historical use in controlling microbial infection. In addition to these screen hits, we investigated and identified imatinib (Gleevec) as an additional, potent modulator of HbF expression and under investigated therapeutic for SCD [34,35]. Our data shows imatinib to exert strong increases in HbF expression in BCR-ABL positive K562 and BCR-ABL negative HEL92.1.7 cell lines. While originally designed as a BCR-ABL specific targeted therapy, it is known to inhibit additional receptor tyrosine kinases, namely platelet-derived growth factor receptor (PDGFR), c-Kit, and c-Abl [36]. Previous work in a Kit-activating mouse model has shown a possible link between continuous Kit activation and partial blockage of erythroid differentiation [31,32]. Furthermore, our 4C data demonstrate shifts in frequency and location of HBG2 promoter *cis* binding interactions concurrent with an increase in expression of HbF. We hypothesize that imatinib may be inhibiting the c-Kit tyrosine kinase and allowing terminal erythroid differentiation signaling to occur in our model cell lines. Further experimentation is needed, however, to characterize the link between imatinib and c-Kit signaling in these cell models. 

Historically there have been very few therapeutic options to treat SCD. Hydroxyurea remained the sole pharmacological agent for almost 20 years before the recent global push for new therapeutics. Given this global effort to find new treatments, we performed a large-scale screen of the NCC to identify new candidates for fast-track use in SCD. Our study is the first to utilize this library and present nine new drugs for consideration for the disease. Furthermore, we suggest pyrimethamine and imatinib as strong candidates for repurposing due to their strong effects on HbF expression, low cytotoxicity, and safe history of use in humans. With further testing and in vivo characterization, these drugs have real potential for combating SCD on a global scale.

## Figures and Tables

**Figure 1 jcm-09-02276-f001:**
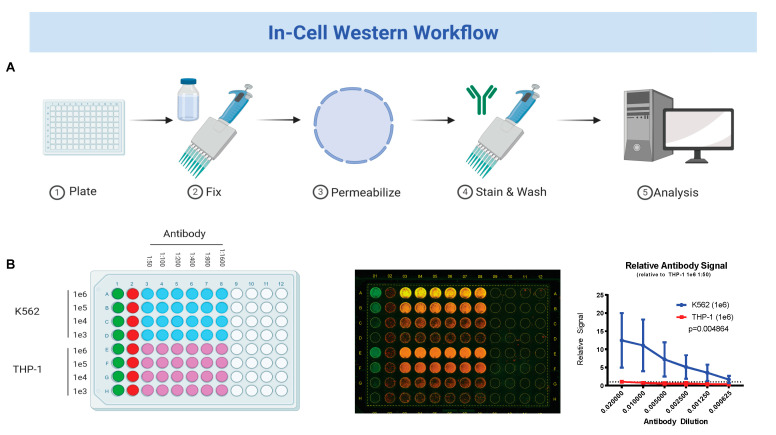
Validation of fetal hemoglobin antibody for in-cell Western workflow. (**A**) Workflow procedure for performing in-cell Western experiment. Further details are available in methods section; (**B**) antibody validation for fetal hemoglobin (HbF) in positive expression cell line (K562) normalized to negative expression cell line (THP-1).

**Figure 2 jcm-09-02276-f002:**
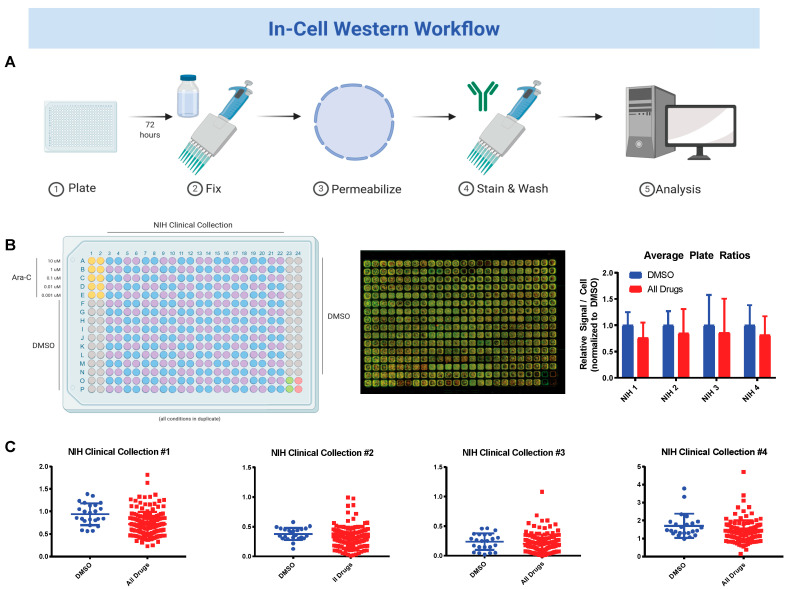
Sequential drug screens show new candidates for sickle-cell disease (SCD). (**A**) Workflow procedure for performing in-cell Western experiment. Further details are available in the methods section; (**B**) sample layout and result for screen of NIH Clinical Collection on K562 cells. Fetal hemoglobin signal normalized to DMSO treated cells; (**C**) signal-to-cell ratios for drug duplicates by plate on K562 cells.

**Figure 3 jcm-09-02276-f003:**
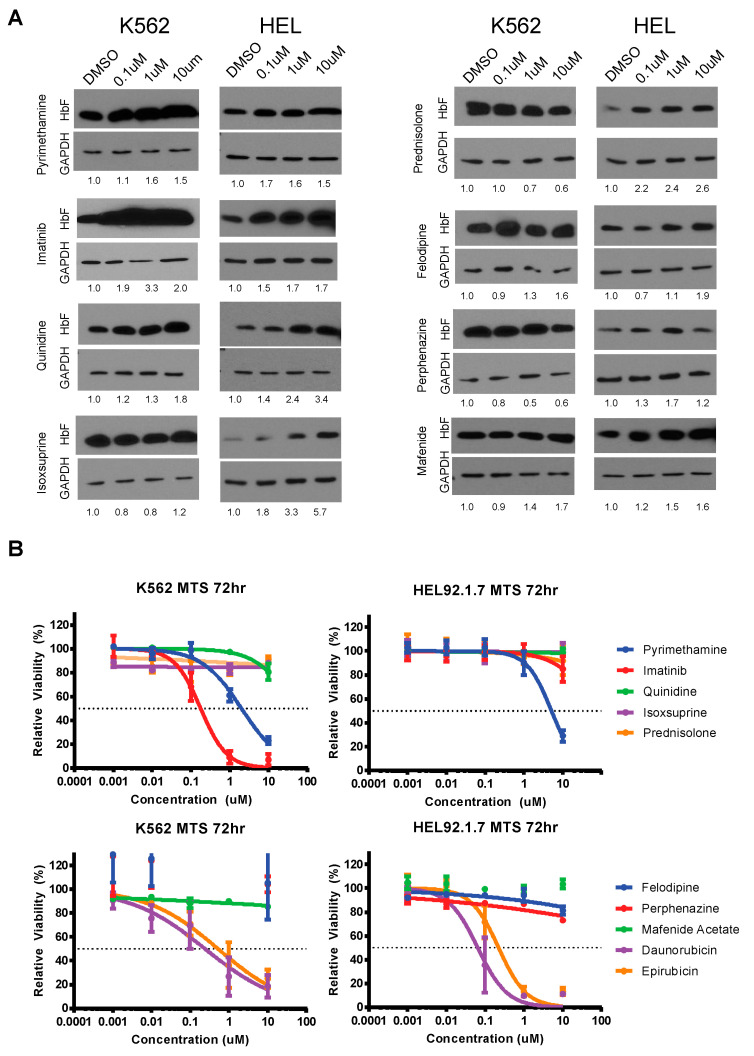
HbF and cytotoxicity profile of overlapping screen hits on K562 and HEL92.1.7 cell lines. (**A**) Immunoblot validations of screen candidates in K562 and HEL92.1.7 cell lines normalized to DMSO; (**B**) MTS viability assays of screen candidates in K562 and HEL92.1.7 normalized to DMSO.

**Figure 4 jcm-09-02276-f004:**
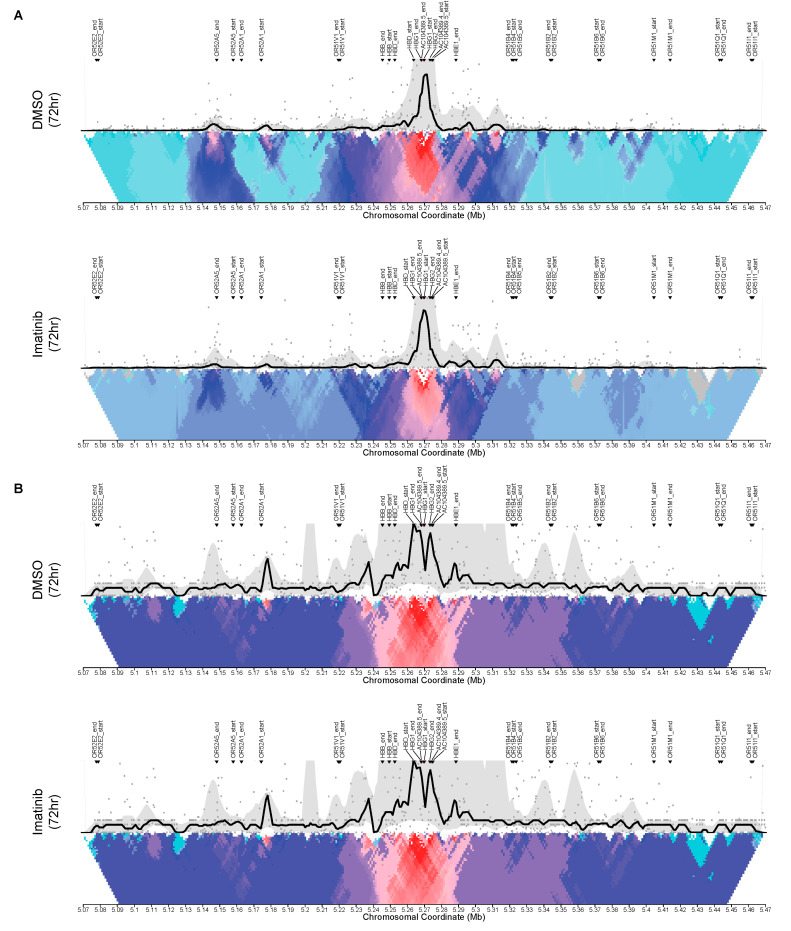
Changes in chromatin arrangement via circular chromosome conformation capture (4c) in K562 and HEL92.1.7 cell lines with imatinib treatment. Libraries were generated using DpnII and Csp61 in subsequent restriction enzyme digestions using primers for the HBG2 promoter (Forward: CAAAGCACCTGGATGATC, Reverse: TTGTCTCTAGCTCCAGTGAG). (**A**) Changes in chromosomal cis-interactions with HBG2 promoter on chromosome 11 with imatinib treatment in K562 cells; (**B**) Changes in chromosomal cis-interactions with HBG2 promoter on chromosome 11 with imatinib treatment in HEL92.1.7 cells.

**Table 1 jcm-09-02276-t001:** Top 20 FDA-approved drugs that positively modulate HbF in K562 s.

Drug Name	Fold Change	Indication
Daunorubicin Hydrochloride	4.55	Acute Myeloid Leukemia
Albendazole	2.86	Anthelmintic
Chloroxine	2.76	Seborrheic Dermatitis
Raltitrexed	2.62	Colorectal Cancer
Epirubicin Hydrochloride	2.58	Breast Cancer
Floxuridine	2.50	Hepatic Cancer
Pindolol	2.50	Hypertension
Methylprednisolone	2.32	Rheumatoid Arthritis
Fexofenadine Hydrochloride	2.28	Seasonal Allergic Rhinitis
6-Azauridine	2.25	Psoriasis
Pyrimethamine	2.23	Malarial Infection
Mafenide Acetate	2.11	Bacterial Infection
Prednisolone	2.09	Congenital Adrenal Hyperplasia
Ribavirin	2.02	Chronic Hepatitis C
Quinidine Hydrochloride	1.99	Cardiac Dysrhythmia
Ebselen	1.95	Type 2 Diabetes Mellitus *
Tegaserod Maleate	1.93	Irritable Bowel Syndrome
Delta1-Hydrocortisone 21-hemisuccinate Sodium	1.90	Corticosteroid-responsive Dematoses
Enalapril Maleate	1.81	Congestive Heart Failure
Granisetron Hydrochloride	1.75	Nausea

* investigational small molecule

**Table 2 jcm-09-02276-t002:** Top 20 FDA-approved drugs that positively modulate HbF in iRBCs (CD45+ cells derived from sickle-cell patients that have been in vitro differentiated towards an erythroid lineage).

Drug Name	Fold Change	Indication
Daunorubicin Hydrochloride	5.76	Acute Myeloid Leukemia
Perphenazine	5.16	Schizophrenia
Acyclovir	5.14	Viral Infection
Cefazolin Sodium	4.71	Bacterial Infection
Epirubicin Hydrochloride	4.59	Breast Cancer
Atenolol	4.49	Hypertension
Methoxsalen	4.44	Cutaneous T-Cell Lymphoma
Ampiroxicam	4.34	Arthritis
Prednisolone	4.32	Rheumatoid Arthritis
Oxytetracycline hydrochloride	4.30	Bacterial Infection
Pfizerpen	4.14	Bacterial Infection
Tetracycline	3.95	Bacterial Infection
Pitavastatin	3.92	Elevated Cholesterol
Benazepril Hydrochloride	3.80	Elevated Blood Pressure
Triamcinolone Acetonide	3.67	Eczema
Phenelzine	3.52	Depression
Digoxin	3.50	Heart Failure
Dibenzyline	3.28	Pheochromocytoma
Buspar	3.28	Anxiety
Fluoxetine Hydrochloride	3.25	Depression

**Table 3 jcm-09-02276-t003:** In-cell Western identifies 9 FDA approved drugs efficacious in K562 s and iRBCs.

Drug Name	K562	iRBC	Indication
Daunorubicin Hydrochloride	4.55	5.76	Acute Myeloid Leukemia
Epirubicin Hydrochloride	2.58	4.59	Breast Cancer
Pyrimethamine	2.23	2.01	Malarial Infection
Mafenide Acetate	2.11	2.71	Bacterial Infection
Prednisolone	2.09	4.32	Immunosuppression
Quinidine Hydrochloride	1.99	2.08	Malarial Infection
Perphenazine	1.71	5.16	Schizophrenia
Felodipine	1.62	1.96	Hypertension
Duvadilan	1.60	2.58	Cerebral Vascular Insufficiency

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
