# Peer review of "Large-Scale Drug Screen Identifies FDA-Approved Drugs for Repurposing in Sickle-Cell Disease"

_jcm, 2020, doi:10.3390/jcm9072276_

Round 1

Reviewer 1 Report

As a screen for inducers of HbF, this seems quite clever, although using K562 cells is fraught, since -once differentiated they can ONLY express HbF, so induction of HbF may mean just erythroid differentiation.  This should be stated, and in future, would consider omitting K562 and focusing on primary cells or HEL cells.

The introduction has many small errors, listed below.

The conclusion has a major error:

We hypothesize that imatinib is inhibiting the c-Kit tyrosine
344 kinase, thus disrupting c-Kit normal signaling cascade and promoting expression of GATA-1 and
 KLF-1. As expression of these transcription factors increases, downstream binding events of each at
the gamma-globin promoter also increase and subsequently gamma-globin expression and HbF
polymer formation increase as well.

They do not show this, and KLF1 is a known REPRESSOR of gamma globin gene expression, and so its induction is not likely to increase gamma globin gene expression.

Might be best to just describe findings and limit speculation, absent data or extensive knowledge of the field.

Quibbles:

VOCs are probably not the sole or major cause of decreased life expectancy, but rather cumulative vasculopathy.

(line 45) Accordingly, sickle cell patients enter a vicious cycle
of VOCs that decrease their overall life expectancy. Despite the severity of the disease and these symptoms. SCD has had very few treatment options to date.

This mechanism does not make sense; Hydrea mostly works by increasing HbF and decreasing WBC:

line 54 [Hydroxyurea...] found to attack SCD pathology at multiple fronts through its ability to first inhibit, then increase the overall kinetics of erythropoiesis and fetal hemoglobin levels

Voxelator increase Hgb, but does not decrease VOCs:

line 58: Voxelotor, formerly
called GBT440, acts a hemoglobin modulator that is able to directly bind to the alpha-globin chain ofHbS polymers and allosterically stabilize the oxygenated state of the molecule, thus inhibiting HbS chain polymerization and drastically reducing the risk of VOC.

Needs editing

line 65: Additionally, the economic costs of recent
therapeutic efforts can be a loft barrier for some (>$2000 per month) [8, 12]. ...lofty barrier...

HbF is not a polymer, and is present throughout fetal life.

Fetal hemoglobin is a naturally occurring hemoglobin polymer that is passingly present during fetal
73 development.

SCD patients that possess the HPFH hereditary trait tend to have a higher quality
77 of life and expression less frequent hospitalizations compared to HPFH negative SCD patients

Reviewer 2 Report

Authors changed some issues, refreshes references in the manuscript but didn't include the following remarks:

1) there were used blood samples in the study, indicate the appropriate number of Bioethical Committee, there should be also indicated in paragraph 2.1. the number of taken blood samples
2) cell lines should be specified, add the type of cell culture (adhesive or suspension), source (leukemia, lymphocytes, etc. - give full name of the cell name, not only acronym)
3) figure 4 is completely not readable, there are overlapping fonts, additionally, the legend description is not sufficient, there should be indicated the source data for this analysis. This analysis is performed only on commercial cell lines, blood samples should be also included for more reliable comparisons.

The human samples issue will be not clear.

Round 2

Reviewer 1 Report

Fine for publication-the most egregious mis-statements have been removed.